# Liver Fibrosis Assessment with Diffusion-Weighted Imaging: Value of Liver Apparent Diffusion Coefficient Normalization Using the Spleen as a Reference Organ

**DOI:** 10.3390/diagnostics9030107

**Published:** 2019-08-28

**Authors:** Min Ki Shin, Ji Soo Song, Seung Bae Hwang, Hong Pil Hwang, Young Jun Kim, Woo Sung Moon

**Affiliations:** 1Department of Radiology, Chonbuk National University Medical School and Hospital, Jeonju 54907, Korea; 2Research Institute of Clinical Medicine of Chonbuk National University, Jeonju 54907, Korea; 3Biomedical Research Institute of Chonbuk National University Hospital, Jeonju 54907, Korea; 4Department of Surgery, Chonbuk National University Medical School, Jeonju 54907, Korea; 5Department of Radiology, Presbyterian Medical Center, Jeonju 54907, Korea; 6Department of Pathology, Chonbuk National University Medical School, Jeonju 54907, Korea

**Keywords:** fibrosis, liver cirrhosis, spleen, diffusion magnetic resonance imaging, magnetic resonance imaging

## Abstract

Liver fibrosis staging is of great clinical importance because it is used to assess the severity of the underlying chronic liver disease. Among various imaging-based methods, apparent diffusion coefficient (ADC) measurement using diffusion-weighted imaging (DWI) has the potential to be used as an imaging biomarker for liver fibrosis assessment. In this study, we investigated the usefulness of liver ADC normalization using the spleen as a reference organ in liver fibrosis staging with 66 patients who underwent liver magnetic resonance imaging (MRI), transient elastography (TE), and surgical resection of a hepatic mass. ADC values of the liver (ADC_liver_) and spleen were analyzed, and the spleen was used for ADC_liver_ normalization (nADC_liver_). ADC_liver_ showed a weak negative correlation with TE (*r* = −0.246; *p* = 0.047) and fibrosis stage (*r* = −0.269; *p* = 0.029), while n ADC_liver_ showed a moderate negative correlation with TE (*r* = −0.504; *p* < 0.001) and fibrosis stage (*r* = −0.579; *p* < 0.001). AUC values for nADC_liver_ (0.777–0.875) were higher than those for ADC_liver_ for each stage of fibrosis (0.596–0.713, *p* = 0.037–0.157). AUC values for TE (0.726–0.884) and nADC_liver_ were not statistically different. In conclusion, normalized liver ADC can be useful in diagnosing liver fibrosis stage in patients with variable DWI acquisitions.

## 1. Introduction

Liver fibrosis is characterized by excessive deposition of extracellular matrix proteins in response to injury and failure of cellular repair efforts to degrade these deposits. Liver fibrosis can progress to cirrhosis, which puts patients at a higher risk for hepatocellular carcinoma (HCC) and hepatic decompensation. As a result, identification of fibrosis precisely over the entire pathologic spectrum, from early-stage fibrosis to cirrhosis, is of considerable clinical importance for managing patients with chronic liver disease [1]. In addition, early detection of fibrosis is important because of the potential to prevent fibrosis progression, or even its reversal by elimination of causative factors [2].

Many noninvasive image-based methods to assess liver fibrosis have been investigated, including transient elastography (TE), magnetic resonance elastography (MRE), diffusion-weighted imaging (DWI), perfusion-weighted imaging, and MR spectroscopy [3,4,5]. Liver stiffness measurements using TE have been widely studied, and TE is efficient in measuring liver stiffness by elastic shear wave propagation through the liver [6,7]. However, its applicability can be limited by patient obesity, the presence of ascites, or limited operator experience [8]. In contrast, DWI can be added easily to a routine MR protocol, providing anatomic and structural information as well as measurement of quantitative metrics, such as the apparent diffusion coefficient (ADC). Many studies have shown that the ADC can be used to detect and stage liver fibrosis [9,10,11]. However, reported ADC values vary between studies due to the influence of many factors, including MR system hardware, patient characteristics, motion artifacts, acquisition parameters, and susceptibility effects [12,13]. Further, to use ADC values as an imaging biomarkers or prognostic parameters in longitudinal or multicenter studies, it is important to standardize ADC measurements.

A previous study of patients undergoing many rounds of MRI at various times with variable acquisition parameters reported that an ADC normalized using the spleen for a reference organ significantly reduced variability in ADC measurements for upper abdominal organs [14]. In order to qualify as a reference, the comparison organ should be similarly affected by diffusion parameters as the examined organ. The spleen could be a suitable reference organ, as it has been shown to have a comparatively non-variable ADC value across different diseases, and its ADC value can be used for quantitative analysis with ratios as needed [15]. In addition, the spleen is often added in liver MR imaging, and is a well-perfused organ.

The purpose of this study was to evaluate the feasibility of using a normalized ADC with the spleen as the reference organ for liver fibrosis assessment by (1) evaluating correlations between TE values and both standard and normalized ADC measurements, respectively, and between pathologic liver fibrosis staging and both standard and normalized ADC measurements, respectively, in patients with variable DWI acquisition parameters and (2) comparing the diagnostic performance of normalized ADC and TE in fibrosis evaluation relative to the gold standard of liver biopsy.

## 2. Materials and Methods

### 2.1. Subjects

This study was approved by institutional review board of the Chonbuk National University Hospital (IRB file No. 2018-02-014-001; approved date, 23 March, 2018) and patient informed consent was waived for reviewing patient images and records. From June 2010 to February 2018, 138 patients undergoing MR imaging of the liver, including DWI, and surgical resection of a hepatic mass were retrospectively reviewed. Of these patients, 55 were excluded for reasons as follows: (1) 40 patients did not have TE measurements; (2) Eight patients had either a very large hepatic mass or multiple masses, which hindered liver ADC measurement; and (3) Seven patients were noted to have severe susceptibility artifacts on DWI. Our final study population of 83 patients included 65 men and 18 women with a mean age of 58.4 years. Of these patients, 76 (91.6%) had chronic hepatitis from the following causes: chronic hepatitis B (*n* = 57), alcoholism (*n* = 10), chronic hepatitis C (*n* = 3), and unknown (*n* = 6).

### 2.2. MR Imaging Techniques

Liver MR imaging was acquired using three 3.0 T MR systems (Verio and Skyra, Siemens Healthineers, Erlangen, Germany; Achieva TX, Philips Healthcare, Best, the Netherlands). Hepatobiliary and dynamic phases were initiated following bolus injection of gadoxetic acid (Eovist or Primovist; Bayer Healthcare, Berlin, Germany). DWI was obtained between the 10- and 20-min hepatobiliary phase.

DWI was executed using either a free-breathing or navigator-triggered method. Various *b* value combinations were used, which included *b*_1_ = 0, 50, 400, 800; *b*_2_ = 0, 50, 600; and *b*_3_ = 50, 400, 800 s/mm^2^. Acquisition parameters are presented in Table 1.

### 2.3. Quantitative Image Analysis

Maps of ADC values were generated automatically with mono-exponential fitting for all *b* values. A radiologist with two years of experience in MR imaging determined regions of interest (ROIs) for all anatomic regions on the ADC images with a picture archiving and communication (PACS) system (Maroview 5.4; Marotech, Seoul, Korea). In accordance with anatomic regions studied and patient characteristics, the ROI sizes were various (range, 2–8 cm^2^). Each ROI was similarly positioned on a corresponding ADC map through the PACS system. The right and left liver lobes and the spleen were the anatomic regions analyzed. For the right and left liver lobes, two ROIs from three contiguous slices were measured, with a central section obtained through the level of the right portal vein for the right lobe and the umbilical portion of the left portal vein for the left lobe. For the spleen, two ROIs from three contiguous slices were determined, with a central section obtained through the level of the splenic hilum. In total, 18 ROIs were determined per patient. For ADC that was non-normalized, the values were absolute; nevertheless, the values are relative and are represented by a ratio with normalized ADC. Normalized ADC may be able to achieve greater standardization across different parameters or different devices, as these differences may be minimized by normalization. Regarding the normalization calculations, the ADC values of liver regions (ADC_liver_) were divided by the spleen ADC values and denoted nADC_liver_. For liver stiffness measurements using TE, a FibroScan system (Echosens, Paris, France) equipped with the standard probe (M-probe) was used as previously described [7,16].

### 2.4. Histopathologic Evaluation

Surgical treatment of the patients included in the final analysis included 30 who underwent segmentectomy, 20 who had sectionectomy, 18 who underwent wedge resection, 10 who had hepatectomy, and five who were treated with liver transplantation. Liver fibrosis stage of all surgical specimens was evaluated by a senior hepatopathologist with more than 23 years of experience using the METAVIR scoring system; stage F0, no fibrosis; stage F1, portal fibrosis; stage F2, periportal fibrosis; stage F3, septal fibrosis; and stage F4, cirrhosis.

### 2.5. Statistical Analysis

The Shapiro-Wilk test was used to assess normality of the continuous variables. Correlations between TE values and liver ADC value (before and after normalization), respectively, and liver fibrosis stage were investigated using Spearman’s rank correlation test. In addition, correlation between liver ADC value (before and after normalization) and TE values were explored using the Pearson correlation test. The area under the receiver operating characteristic (ROC) curve (AUC value) and the optimal cut-off value were calculated for differentiating fibrosis stages ≥F1 from F0, ≥F2 from ≤F1, ≥F3 from ≤F2, and F4 from ≤F3. The cut-off was determined by the maximum sum of sensitivity and specificity values. AUC values were compared using the jackknife method [17]. Statistical analysis was performed using commercially available software (SPSS v23.0; IBM, Armonk, NY, USA or MedCalc v13.0.0.0; MedCalc Software, Ostend, Belgium). *p* values lower than 0.05 were considered significant.

## 3. Results

### 3.1. DWI Acquisition Methods

For the 83 examinations in our patient population, the MR system used most was Verio (45 examinations, 54.2%), then Achieva (20 examinations, 24.1%), and then Skyra (18 examinations, 21.7%). Of the two respiratory motion compensation methods, free breathing was used more frequently than navigator triggering (72.7% vs. 27.3%). For *b* value parameters, a *b* value combination of 0, 50, 400, and 800 (*b*_1_) was used most (48 examinations, 72.7%) (Table 2).

### 3.2. Histopathologic Results

Based on the METAVIR scoring system, the following fibrosis stage distribution was observed: stage F0 (*n* = 13), stage F1 (*n* = 8), stage F2 (*n* = 11), stage F3 (*n* = 26), and stage F4 (cirrhosis, *n* = 25) (Figure 1).

### 3.3. Correlations Between Fibrosis Stage, TE Value, Liver ADC and Normalized Liver ADC

We observed a weak negative correlation between TE values and ADC_liver_ (*r* = −0.256, [95% CI: −0.450–0.014]; *p* = 0.045), while a moderate negative correlation was noted between TE values and nADC_liver_ (*r* = −0.523, [95% CI: −0.647–0.275]; *p* < 0.001). In addition, there was a weak negative correlation between fibrosis stage and ADC_liver_
*(r* = −0.281 [95% CI: −0.471–0.018]; *p* = 0.026), while a moderate negative correlation was observed between fibrosis stage and nADC_liver_ (*r* = −0.584 [95% CI: −0.703–0.382]; *p* < 0.001) (Figure 2). Finally, we observed a moderate positive correlation between fibrosis stage and TE values (*r* = 0.665, [95% CI: 0.502–0.771]; *p* < 0.001).

### 3.4. Diagnostic Performance and Cut-off Value Evaluation

According to ROC analysis, nADC_liver_ exhibited a good diagnostic performance for each stage of fibrosis, with AUCs higher than those of ADC_liver_. Comparing nADC_liver_ vs ADC_liver_, our results were as follows: (1) for fibrosis stage ≥F1, 0.863 vs 0.625 (*p* = 0.064); (2) for fibrosis stage ≥F2, 0.877 vs 0.631 (*p* = 0.031); (3) for fibrosis stage ≥F3, 0.764 vs 0.587 (*p* = 0.114); and (4) for fibrosis stage F4, 0.789 vs 0.577 (*p* = 0.041). The optimal cut-off values of nADC_liver_ were 1.443, 1.411, 1.396, and 1.365 for diagnosing ≥F1, ≥F2, ≥F3, and F4, respectively (Table 3) (Figure 3). AUC values of TE for diagnosing ≥F1, ≥F2, ≥F3, and F4 were 0.799, 0.811, 0.721, and 0.884 respectively, none of which were significantly different from comparable values for nADC_liver_ (*p* > 0.05) (Table 4) (Figure 4).

## 4. Discussion

In this study, we observed moderate negative correlations between nADC_liver_ and both TE values and fibrosis stage (*r* = −0.523 and −0.584, respectively). ADC_liver_ was negatively correlated with both TE values and fibrosis stage as well, although it was only weakly correlated with both (*r* = −0.256 and −0.281, respectively). Based on the results of AUC, the diagnostic performance of nADC_liver_ was superior to ADC_liver_, with significant differences in fibrosis staging results for ≥F2 (*p* = 0.031) and ≥F4 (*p* = 0.041). nADC_liver_ demonstrated good diagnostic performance in diagnosing all stages of fibrosis in comparison to TE (*p* > 0.05, no significant differences between the parameters).

Previous studies have reported ADC values of patients with liver fibrosis are significantly lower than those of normal controls and ADC values decrease as the degree of fibrosis progresses [9,10]. A possible explanation for this phenomenon is that since fibrotic liver tissue has abundant proton-poor connective tissue, both blood flow and water diffusion are restricted, which results in a decrease in its ADC values [18]. Although liver biopsy is often contraindicated in patients with advanced cirrhosis, due to the presence of ascites, coagulopathy, and other co-morbidities, it is still regarded as the gold standard for liver fibrosis staging. The use of DWI in patients with contraindications for liver biopsy has significant clinical impact due to its wide availability and relative safety [11]. Nevertheless, a lack of standardized techniques for DWI acquisition is a significant drawback when using ADC as an imaging biomarker for longitudinal multicenter studies, as reported ADC values vary widely with considerable overlap between normal and abnormal ranges. In addition, good reproducibility of ADC measurements is necessary, along with optimization and standardization of techniques for DWI acquisition. Previous studies have reported that reasonable reproducibility levels can be obtained with ADC measurements of the upper abdomen for patients and volunteers [19,20]. In the current study, with the use of variable *b* value combinations, the optimal cut-off value for diagnosing cirrhosis (F4) was 1.189 × 10^−3^ mm^2^/s, with an AUC value of 0.577. When using nADC_liver_, the AUC value increased to 0.789, which was significantly higher than that of ADC_liver_ (*p* = 0.041). We theorize that since most of our DWI studies (73.5%) were performed with *b* values of 0, 50, 400, and 800 s/mm^2^, this fact may explain both the correlation between ADC_liver_ and pathologic fibrosis and the somewhat comparable diagnostic performance of ADC_liver_ and nADC_liver_.

It is widely accepted that significant fibrosis (F2) is a predictor of future liver cirrhosis and that the ultimate goal of treatment at this stage is to cure the patient by eliminating the underlying cause of liver disease. In addition, maximum accuracy in the detection of advanced fibrosis (F3) or cirrhosis (F4) is important, since these patients should be screened for portal hypertension and HCC [21]. Our study revealed significantly higher AUC values of nADC_liver_ for diagnosing significant fibrosis (≥F2, *p* = 0.031) and cirrhosis (F4, *p* = 0.041) compared to ADC_liver_; and these values were comparable to those in previous studies without normalization of ADC, which showed AUC values of 0.730–0.935 for the detection of cirrhosis [3,15,22]. However, this is the first study to use variable DWI acquisition parameters as well as different MR systems to evaluate liver fibrosis. It demonstrated that normalization of ADC values, using the spleen as the reference organ, significantly increased their diagnostic performance for diagnosing ≥F2 and F4.

Transient elastography (TE) has been studied in large patient cohorts for liver fibrosis and cirrhosis detection [6,23]. Interestingly, there was no significant difference in diagnostic performance between TE and nADC_liver_ in relation to fibrosis. Although TE measures the stiffness of a liver parenchymal volume 100 times bigger than that of a liver biopsy, it cannot enable segmental tissue analysis or be performed in patients with ascites or obesity [16]. In contrast, DWI includes the whole liver volume with the capability of making ADC measurements for each liver segment, thus providing information about the most severely affected liver segment, and can be performed even in obese patients or patients with ascites [9]. In addition, the multi-parametric nature of MRI itself allows more complete assessment of organ structure and function, such as the ability to quantify hepatic fat and iron content or the use of dynamic contrast enhancement for hepatic perfusion quantification and HCC detection [9]. Based on our study results, MRI may be of value even in patients without previous TE results, since the progression of liver fibrosis can be assessed using nADC_liver_, even if the data was acquired with different MR systems or acquisition parameters.

This study has several limitations. First, due to the retrospective nature of the study, it has an inherent selection bias. Second, we studied a heterogeneous patient population with different causes of chronic liver disease and had a limited number of patients with early (F1) and significant (F2) fibrosis. Third, although we used resected liver specimens to stage liver fibrosis, it is well known that histopathologic assessment can have high inter- and intraobserver variability, and the reproducibility of these observations could not be examined [2]. Fourth, most of DWI was acquired with *b* values of 0, 50, 400, and 800 s/mm^2^ (73.5%). Fifth, DWI was acquired after gadoxetic acid was administered, but any effects of gadoxetic acid while DWI is performed can be dismissed for up to 20 min following injection [24]. Sixth, since we included *b* values below 150 s/mm^2^ which reflects capillary perfusion, our ADC values were biased. Further study using *b* values above 150 s/mm^2^ is warranted.

In conclusion, our study validated the hypothesis that liver ADC normalized using the spleen as a reference has a better diagnostic performance for detecting stages of fibrosis than liver ADC in patients with variable DWI acquisitions. Further, DWI with normalized liver ADC exhibited comparable diagnostic performance to TE. Future prospective studies with larger patient cohorts and more variation in acquisition parameters and MR systems are necessary to confirm our results and promote application of normalized ADC values.

## Figures and Tables

**Figure 1 diagnostics-09-00107-f001:**
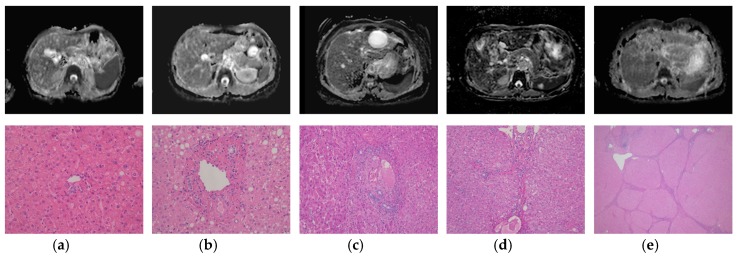
Magnetic resonance imaging (apparent diffusion coefficient (ADC) map, upper row) and histologic findings (H&E stain, lower row) matched for each stages of liver fibrosis. (**a**) No liver fibrosis (F0), magnification ×400. ADC_liver_ was 1.389 × 10^−3^ mm^2^/s and nADC_liver_ was 1.977. (**b**) Portal fibrosis (F1), magnification ×400. ADC_liver_ was 1.091 × 10^−3^ mm^2^/s and nADC_liver_ was 1.327. (**c**) Periportal fibrosis (F2), magnification ×400. ADC_liver_ was 1.376 × 10^−3^ mm^2^/s and nADC_liver_ was 1.416. (**d**) Septal fibrosis (F3), magnification ×200. ADC_liver_ was 0.963 × 10^−3^ mm^2^/s and nADC_liver_ was 1.29. (**e**) Cirrhosis (F4), magnification ×20. ADC_liver_ was 1.278 × 10^−3^ mm^2^/s and nADC_liver_ was 1.355.

**Figure 2 diagnostics-09-00107-f002:**
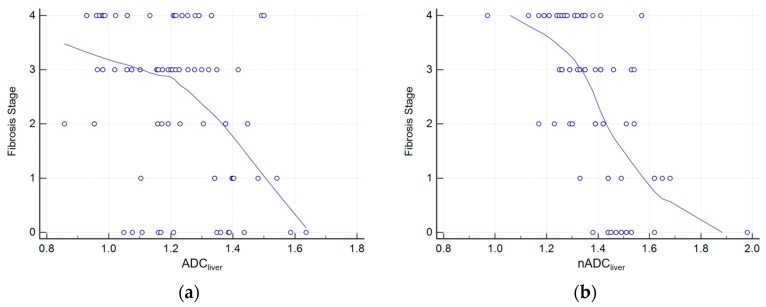
Scatter plot of Spearman’s rank correlation test between liver fibrosis stage and liver ADC before (**a**) and after (**b**) normalization.

**Figure 3 diagnostics-09-00107-f003:**
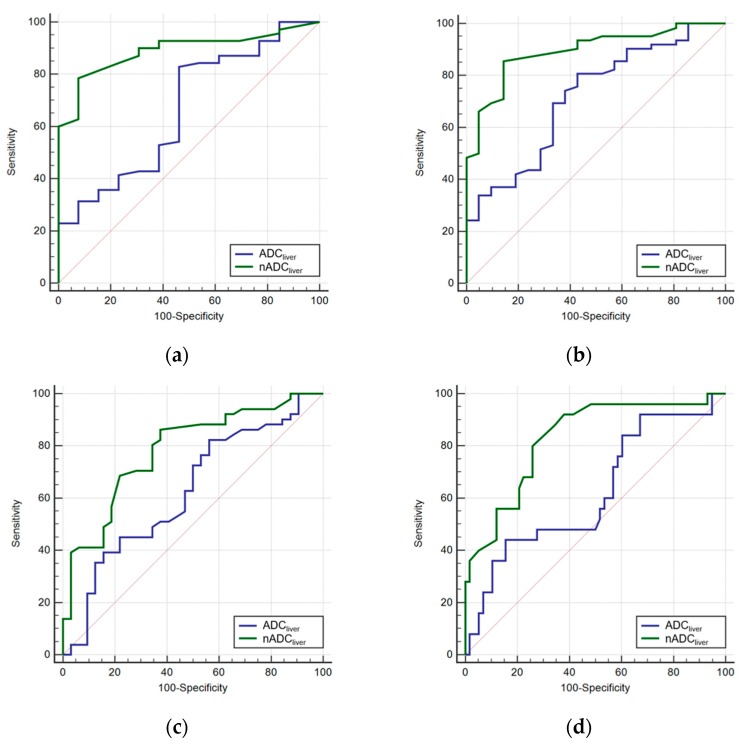
Comparison of receiver operating characteristics curve for diagnosing fibrosis stage ≥F1 (**a**), ≥F2 (**b**), ≥F3 (**c**), and F4 (**d**) on liver ADC (ADC_liver_) and normalized ADC (nADC_liver_)**.**

**Figure 4 diagnostics-09-00107-f004:**
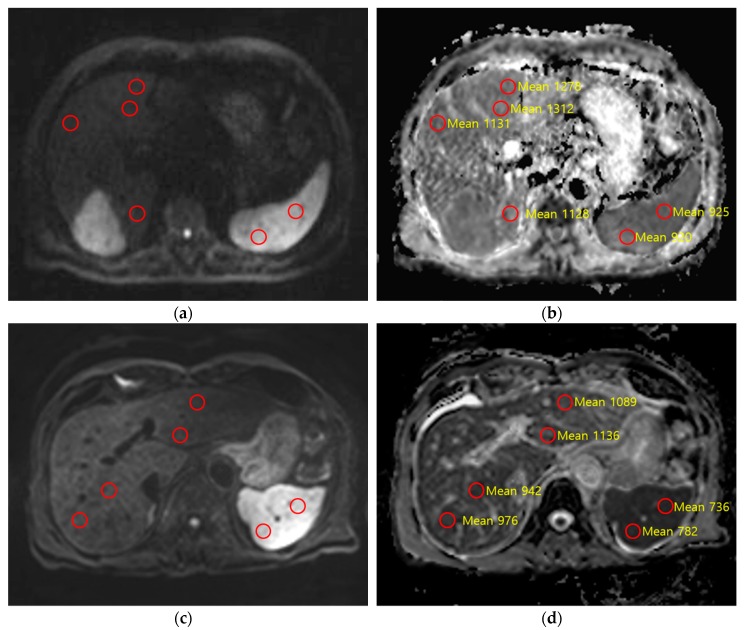
Two patients who underwent surgical resection for hepatocellular carcinoma (HCC) and pathologically confirmed cirrhosis (F4). Red circle indicates region of interest for ADC value measurement (yellow numbers in (**b**,**d**)). (**a**,**b**) A 77-year old man with viral B cirrhosis, navigator-triggered DWI acquired on Verio with *b* = 50, 400, 800 s/mm^2^. On *b* = 800 s/mm^2^ (**a**) and corresponding ADC map (**b**), the liver ADC was 1.236 × 10^−3^ mm^2^/s and the normalized ADC was 1.257. The transient elastography value was 13.2 kPa. (**c**,**d**) A 68-year old man with viral B cirrhosis. Navigator-triggered DWI acquired on Achieva with *b* = 0, 50, 400, 800 s/mm^2^. On *b* = 800 s/mm^2^ (**c**) and corresponding ADC map (**d**), the liver ADC was 0.970 × 10^−3^ mm^2^/s and the normalized ADC was 1.276. The transient elastography value was 12.3 kPa.

**Table 1 diagnostics-09-00107-t001:** Parameters for diffusion-weighted imaging (DWI) acquisition.

Parameter	Verio		Achieva		Skyra
Sequence	SE-EPI	SE-EPI	SE-EPI	SE-EPI	SE-EPI
Respiration	FB	NT	FB	NT	FB
TR/TE (msec)	11500/67	2800/65	8750/66	1422/56	5100/66
FOV (mm)	400 × 400	380 × 285	400 × 400	350 × 350	370 × 278
Matrix	128 × 128	128 × 96	128 × 128	128 × 124	128 × 96
ST (mm)	5	6	5	6	5
Intersection gap (mm)	1	1.2	1	1	1
No. of sections	33	29	35	35	34
NSA	3	2	3	3	4
*b* values (s/mm^2^)	*b*_1_, *b*_3_	*b*_1_, *b*_3_	*b*_1_, *b*_2_	*b* _1_	*b* _1_
BW (Hz)	2442	2298	3743	3634	2442
PAF	GRAPPA = 2	GRAPPA = 2	SENSE = 2	SENSE = 2	GRAPPA = 2
Fat saturation	SPAIR^1^	SPAIR^1^	SPAIR^2^	SPAIR^2^	SPAIR^1^
Scan time	5:50	5:15	5:35	4:00	3:50
EPI factor	96	96	65	65	96

TR = repetition time, TE = echo time, FOV = field of view, NT = navigator-triggered, SE-EPI = spin echo-echo planar imaging, FB = free breathing, PAF = parallel acquisition factor, NSA = number of signals acquired, BW = bandwidth, GRAPPA = generalized autocalibrating partially parallel acquisition, SPAIR^1^ = spectrally adiabatic inversion recovery, ST = slice thickness, SPAIR^2^ = spectral attenuated inversion recovery, SENSE = sensitivity encoding, *b*_1_ = 0, 50, 400, 800; *b*_2_ = 0, 50, 600; *b*_3_ = 50, 400,800 s/mm^2^.

**Table 2 diagnostics-09-00107-t002:** DWI acquisition methods.

*b* Value	Verio		Achieva		Skyra		No. of Examinations
	FB	NT	FB	NT	FB	NT	
*b* _1_	26	2	12	3	18		61 (73.5%)
*b* _2_			4				4 (4.8%)
*b* _3_	3	15					18 (21.7%)
	46 (55.4%)		19 (22.9%)		18 (21.7%)		83 (100%)
	NT = 20 (24.1%), FB = 63 (75.9%)						

FB = free breathing, NT = navigator-triggered, *b*_1_ = 0, 50, 400, 800; *b*_2_ = 0, 50, 600; *b*_3_ = 50, 400,800 s/mm^2^.

**Table 3 diagnostics-09-00107-t003:** Comparison between liver ADC and nADC_liver._

Variable	Liver ADC	nADC_liver_	*p* Value
F1 (*n* = 8)			
Optimal cut-off value	1.347 (×10^−3^ mm^2^/s)	1.443	
Sensitivity (%)	83.1 (71.2, 91.7)	78.2 (64.7, 89.1)	
Specificity (%)	57.2 (24.3, 85.3)	91.0 (67.6, 99.2)	
AUC (95% CI)	0.625 (0.501, 0.727)	0.863 (0.755, 0.952)	0.064
F2 (*n* = 11)			
Optimal cut-off value	1.332 (×10^−3^ mm^2^/s)	1.411	
Sensitivity (%)	83.4 (71.8, 90.6)	84.3 (71.1, 91.9)	
Specificity (%)	58.5 (30.7, 79.9)	86.9 (61.2, 98.1)	
AUC (95% CI)	0.631 (0.529, 0.759)	0.877 (0.772, 0.948)	0.031
F3 (*n* = 26)			
Optimal cut-off value	1.330 (×10^−3^ mm^2^/s)	1.396	
Sensitivity (%)	85.4 (71.2, 92.1)	84.5 (69.8, 92.1)	
Specificity (%)	44.2 (23.6, 65.1)	69.2 (46.9, 85.8)	
AUC (95% CI)	0.587 (0.461, 0.694)	0.764 (0.645, 0.859)	0.114
F4 (*n* = 25)			
Optimal cut-off value	1.189 (×10^−3^ mm^2^/s)	1.365	
Sensitivity (%)	43.4 (22.1, 65.7)	90.2 (68.2, 98.9)	
Specificity (%)	83.1 (68.5, 90.8)	62.3 (46.6, 75.8)	
AUC (95% CI)	0.577 (0.443, 0.689)	0.789 (0.671, 0.882)	0.041

Note. 95% confidence intervals are given in parenthesis for sensitivity, specificity, and AUC.

**Table 4 diagnostics-09-00107-t004:** Comparison between TE and nADC_liver._

Variable	TE	nADC_liver_	*p* Value
F1 (*n* = 8)			
Optimal cut-off value	5.9 (kPa)	1.443	
Sensitivity (%)	94.1 (84.7, 98.8)	78.2 (64.7, 89.1)	
Specificity (%)	58.2 (26.2, 88.2)	91.0 (67.6, 99.2)	
AUC (95% CI)	0.799 (0.683, 0.888)	0.863 (0.755, 0.952)	0.612
F2 (*n* = 11)			
Optimal cut-off value	6.9 (kPa)	1.411	
Sensitivity (%)	88.2 (75.8, 95.3)	84.3 (71.1, 91.9)	
Specificity (%)	68.8 (41.5, 89.2)	86.9 (61.2, 98.1)	
AUC (95% CI)	0.811 (0.718, 0.882)	0.877 (0.772, 0.948)	0.892
F3 (*n* = 26)			
Optimal cut-off value	9.0 (kPa)	1.396	
Sensitivity (%)	65.1 (49.0, 77.9)	84.5 (69.8, 92.1)	
Specificity (%)	71.2 (50.2, 87.1)	69.2 (46.9, 85.8)	
AUC (95% CI)	0.721 (0.597, 0.802)	0.764 (0.645, 0.859)	0.877
F4 (*n* = 25)			
Optimal cut-off value	9.7 (kPa)	1.365	
Sensitivity (%)	100 (83.2, 100)	90.2 (68.2, 98.9)	
Specificity (%)	69.6 (54.2, 82.3)	62.3 (46.6, 75.8)	
AUC (95% CI)	0.884 (0.787, 0.943)	0.789 (0.671, 0.882)	0.064

Note. 95% confidence intervals are given in parenthesis for sensitivity, specificity, and AUC.

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
