# Peer review of "Liver Fibrosis Assessment with Diffusion-Weighted Imaging: Value of Liver Apparent Diffusion Coefficient Normalization Using the Spleen as a Reference Organ"

_diagnostics, 2019, doi:10.3390/diagnostics9030107_

Round 1

Reviewer 1 Report

This paper reports on a study to use apparent diffusion coefficients (ADC's) normalised by the ADC of the spleen. It is found that diagnostic relevance (AUC) is improved as compared to not normalised ADC's. It was further found that AUC's are comparable to transient elastography (TE), thus normalised ADC appear to be a viable alternative method to asses and monitor liver fibrosis. The authors do not propose a new method (and do not claim to do so) but benchmark this method in a field study.

While the manuscript is written in a accessible way I was not sure how evidence is provided to the authors claim. There is no link provided between the numbers listed in Table 3 and 4 and the ADC's supposedly extracted from diffusion weighted images at different b values. All we see are two images and two ADC maps. No figures are provided which would demonstrate how monoexponential fits are applied and ADC's are extracted for characteristic and non-characteristic voxels. No trend is shown for (n)ADC's  vs. echo time or fibrosis state. No examples of receiver operating characteristic curves are shown. So there is nothing in the manuscript which would give me any reason to assume that numbers in Table 3 and 4 are correct. Under this circumstances I can not recommend publication.

Author Response

R1-1. While the manuscript is written in a accessible way I was not sure how evidence is provided to the authors claim. There is no link provided between the numbers listed in Table 3 and 4 and the ADC's supposedly extracted from diffusion weighted images at different b values. All we see are two images and two ADC maps. No figures are provided which would demonstrate how monoexponential fits are applied and ADC's are extracted for characteristic and non-characteristic voxels.

Answer) Thank you for your comment. We’ve added ROIs and its mean values in the figure in order to make it more clear how it was measured and calculated (Fig. 3).

R1-2. No trend is shown for (n)ADC's echo time or fibrosis state. No examples of receiver operating characteristic curves are shown. So there is nothing in the manuscript which would give me any reason to assume that numbers in Table 3 and 4 are correct.

Answer) We appreciate your comment. We’ve added a scatter plot of Spearman’s rank correlation between liver ADC (before and after normalization) and liver fibrosis stage (Fig. 1). In addition, we’ve added a figure of comparing ROC curves of ADCliver and nADCliver for all fibrosis stages (Fig. 2).

Reviewer 2 Report

Dear authors,

congratulations on the nice manuscript. I have few comments prior recommendation for acceptance.

1) You did not correct the ADC for perfusion effects. As you know, b-values below 150 are significantly influenced by perfusion, thus, your ADC is biased. If you only use b-values abvoe 150 for calculation of the ADC, pure diffusion is reflected. Please amend the limitations section.

2) Did you check for normal distribution of values prior significance testing? I recommend Shapiro-Wilks-Test as relevant amendment.

3) Discussion needs more focus and is too long for my taste, please shorten.

Author Response

R2-1. You did not correct the ADC for perfusion effects. As you know, b-values below 150 are significantly influenced by perfusion, thus, your ADC is biased. If you only use b-values abvoe 150 for calculation of the ADC, pure diffusion is reflected. Please amend the limitations section.
Answer) We appreciate your comment. In accordance with your comment, we’ve added the following sentence as another limitation of our study in the discussion section:
‘Sixth, since we included b values below 150 s/mm2 which reflects capillary perfusion, our ADC values were biased. Further study using b values above 150 s/mm2 is warrented.’

R2-2. Did you check for normal distribution of values prior significance testing? I recommend Shapiro-Wilks-Test as relevant amendment.
Answer) We appreciate your comment. Yes, we’ve checked normality using Shapiro-Wilks test and added a sentence in the statistical analysis section of the manuscript.

R2-3. Discussion needs more focus and is too long for my taste, please shorten.

Answer) We appreciate your comment. We have removed few sentences from the discussion section as appropriate.

Round 2

Reviewer 1 Report

My concerns have been addressed. There would be still room for improvments (such as plots of nADC's vs echo time or fibrosis stage but the authors do provide sufficient evidence for their claims.

Author Response

Thank you very much for your kind comment.